# Ion Channel Properties of a Cation Channelrhodopsin, *Gt*_CCR4

**Shunta Shigemura [1], Shoko Hososhima [1], Hideki Kandori [1,2] and Satoshi P. Tsunoda [1,3,\*]**

[1]   Department of Life Science and Applied Chemistry, Nagoya Institute of Technology, Showa-ku, Nagoya 466-8555, Japan

[2]   OptoBio Technology Research Center, Nagoya Institute of Technology, Showa-ku, Nagoya 466-8555, Japan

[3]   PRESTO, Japan Science and Technology Agency, 4-1-8 Honcho, Kawaguchi, Saitama 332-0012, Japan

\*   Correspondence: tsunoda.satoshi@nitech.ac.jp; Tel.: +81-52-735-5218

**Abstract:** We previously reported a cation channelrhodopsin, *Gt*_CCR4, which is one of the 44 types of microbial rhodopsins from a cryptophyte flagellate, *Guillardia theta*. Due to the modest homology of amino acid sequences with a chlorophyte channelrhodopsin such as *Cr*_ChR2 from *Chlamydomonas reinhardtii*, it has been proposed that a family of cryptophyte channelrhodopsin, including *Gt*_CCR4, has a distinct molecular mechanism for channel gating and ion permeation. In this study, we compared the photocurrent properties, cation selectivity and kinetics between well-known *Cr*_ChR2 and *Gt*_CCR4 by a conventional path clamp method. Large and stable light-induced cation conduction by *Gt*_CCR4 at the maximum absorbing wavelength (530 nm) was observed with only small inactivation (15%), whereas the photocurrent of *Cr*_ChR2 exhibited significant inactivation (50%) and desensitization. The light sensitivity of *Gt*_CCR4 was higher ($EC_{50}$ = 0.13 mW/mm$^2$) than that of *Cr*_ChR2 ($EC_{50}$ = 0.80 mW/mm$^2$) while the channel open life time (photocycle speed) was in the same range as that of *Cr*_ChR2 (25~30 ms for *Gt*_CCR4 and 10~15 ms for *Cr*_ChR2). This observation implies that *Gt*_CCR4 enables optical neuronal spiking with weak light in high temporal resolution when applied in neuroscience. Furthermore, we demonstrated high Na$^+$ selectivity of *Gt*_CCR4 in which the selectivity ratio for Na$^+$ was 37-fold larger than that for *Cr*_ChR2, which primarily conducts H$^+$. On the other hand, *Gt*_CCR4 conducted almost no H$^+$ and no Ca$^{2+}$ under physiological conditions. These results suggest that ion selectivity in *Gt*_CCR4 is distinct from that in *Cr*_ChR2. In addition, a unique red-absorbing and stable intermediate in the photocycle was observed, indicating a photochromic property of *Gt*_CCR4.

**Keywords:** microbial rhodopsin; channelrhodopsin; electrophysiology; optogenetics

## 1. Introduction

Microbial-type rhodopsins are made up of seven or eight transmembrane helices with a covalently bound all-*trans* retinal as the chromophore [1]. They are found in archaea, bacteria, eukaryota (such as fungi and algae) and viruses, and are physiologically responsible for energy production and the phototaxis reaction. Molecular functions of microbial rhodopsin involve ion transporters, sensors and light-regulated enzymes. As for ion-transporting rhodopsins, they are divided into ion-pumps and channels. Bacteriorhodopsin (BR) was the first identified outward directed proton-pumping rhodopsin [2]. The discovery of a Cl$^-$ pump, an Na$^+$ pump and inward-directed proton pumps has been even until recently [3–6]. Structure-based and spectroscopic studies, when combined with electrophysiology and molecular dynamics studies, revealed the detailed molecular mechanism of bacteriorhodopsin and other pumps.

Channelrhodopsin-1 and -2 (*Cr*_ChR1 and *Cr*_ChR2) from *Chlamydomonas reinhardtii* were the first light-gated ion channels to be discovered [7,8]. These homologous proteins permeate cations in which the permeability ratio of $H^+$, $Na^+$, and $K^+$ is $10^6$, 1, and 0.5, respectively. High-resolution X-ray structures revealed details of their molecular architecture and provided insight into their photoactivation and ion conduction [9,10].

Successful expression of *Cr*_ChR2 in neurons allowed the action potential to be manipulated by light, which opened up a new field of research, optogenetics [11,12]. A number of variant molecules have been engineered to improve the functionality of ChR, and homologous ChRs were then reported [13]. Color-tuning variants cover almost the entire visible range. *Cr*_ChR2 displays an action spectrum maximum at 470 nm [8]. ChR variants such as C1V1, which is the chimeric version of ChR1 from *Chlamydomonas reinhardtii* and *Volvox carteri*, or C1C2 (a green receiver) absorb light at around 530~545 nm [14–16]. Another red-shifted ChR, Chrimson from *Chlamydomonas noctigama*, exhibits an absorption maximum at 590 nm which allows reliable neuronal stimulation by light exceeds 600 nm [17]. On the other hand, *Ts*ChR or *Ps*ChR absorb a shorter wavelength, making it possible to excite neurons at 440 nm [18].

The lifetime of an open channel can be extended by mutations at C128 and D156 (DC pair) which form a hydrogen bond bridge in *Cr*_ChR2. Mutations at C128 to Thr, Ala and Ser slowed the kinetics of channel closing 200, 5000 and 10,000-fold respectively [19]. *Cr*_ChR2 D156C displayed an even stronger effect, namely higher light sensitivity and prolonged lifetime of the open channel, by as much as 30 min [20].

Converting ion selectivity is challenging. The *Cr*_ChR2 L132C mutant showed improved $Ca^{2+}$ permeability [21]. The permeability ratio between $H^+$ and $Na^+$ could be modified by a replacement at E143 to A in Chrimson [22]. Anion channelrhodopsins were engineered or discovered from nature and they have been applied as neuronal silencing tools [23–25]. The crystal structures of anion channelrhodopsins revealed their unique features related to the channel gating mechanism [26–28].

A novel cation channelrhodopsin family was reported in 2016 and 2017 from *Guillardia theta*, namely *Gt*_CCR1-4 [29,30]. These cation channelrhodopsins (CCRs) from cryptophyte algae are more homologous to haloarchaeal rhodopsins, such as proton pumping bacteriorhodopsin, than to chlorophyte CCRs, including *Cr*_ChR2. Actually, *Gt*_CCRs conserve the characteristic amino-acid residues involved in unidirectional proton transfer, including the proton acceptor D85 and the proton donor D96 in bacteriorhodopsin (Table S1).

On the other hand, a characteristic glutamic acid in TM2 (E90 in *Cr*_ChR2) which is crucial for channel gating and ion selectivity, is not conserved in *Gt*_CCRs [23,31]. *Cr*_ChR2 possesses a so-called DC pair (C128 and D156 in *Cr*_ChR2), which is responsible for the channel life time [19,32,33]. This is not found in *Gt*_CCRs. Thus, overall sequence patterns separate these cryptophyte CCRs form chlorophyte channels. The molecular mechanisms such as channel gating mechanism and ion selectivity could be distinct in chlorophyte CCRs. Sineshchekov and coworkers already revealed that the retinal Schiff-base (SB) in *Gt*_CCR2 rapidly deprotonates to the D85 homolog, as in BR, upon photoisomerization [34]. Channel-opening requires deprotonation of the D96 homolog. We independently identified photocycle intermediates during the channel function of *Gt*_CCR4 from electrophysiological and flash photolysis experiments. The M-decay corresponds to channel-closing, implicating tight coupling between retinal dynamics and channel function. However, reprotonation of SB for channel closing was achieved by the direct return of a proton from the D85 homolog. Such proton transfers are not the case with *Cr*_ChR2. In *Cr*_ChR2, D156 in TM4 provides the proton [35]. We demonstrated, using an FTIR study, that the secondary structural change in the primary reaction was much smaller than in *Cr*_ChR2 [30]. These differences in the molecular mechanism place the cryptophyte CCR in a new family of channelrhodopsins, which we described as "DTD channelrhodopsins" or "BR-like cation channelrhodopsins" [29,30]. To further reveal the characteristics of these DTD channelrhodopsins, in this study we performed electrophysiological measurements in parallel with *Cr*_ChR2.

## 2. Materials and Methods

### 2.1. Expression Plasmids

A mammalian expression plasmid peGFP-*Gt*_CCR4 was described previously [30]. pVenus-N1-Chop2-315 was a kind gift from Prof. Yawo (The University of Tokyo) [12].

### 2.2. Cell Culture

The electrophysiological assays of *Gt*_CCR4 and *Cr*_ChR2 were performed on ND7/23 cells, which are hybrid cell lines derived from neonatal rat dorsal root ganglia neurons fused with mouse neuroblastoma [36]. ND7/23 cells were grown on a collagen-coated coverslip in Dulbecco's modified Eagle's medium (Wako, Osaka, Japan) supplemented with 2.0 μM all-*trans* retinal and 5% fetal bovine serum, and under a 5% $CO_2$ atmosphere at 37 °C. The expression plasmids were transiently transfected by using the FuGENE HD transfection Reagent (Promega, Fitchburg, WI, USA) according to the manufacturer's instructions. Electrophysiological recordings were then conducted 24–36 h after transfection. Successfully transfected cells were identified by eGFP or Venus fluorescence under a microscope prior to the measurements.

### 2.3. Electrophysiology

All experiments were carried out at room temperature (22 ± 2 °C). Photocurrents were recorded as previously described using an Axopatch 200B amplifier (Molecular Devices, Sunnyvale, CA, USA) under a whole-cell patch clamp configuration [12]. Data were filtered at 5 kHz and sampled at 20 kHz (Digdata1550, Molecular Devices, Sunnyvale, CA, USA) and stored in a computer (pClamp10.6, Molecular Devices). Pipette resistance was 3–6 MΩ. The standard internal pipette solution for the whole-cell voltage clamp contained (in mM) 120 KOH, 100 glutamate, 2.5 $MgCl_2$, 2.5 MgATP, 0.01 Alexa568, 50 HEPES, and 5 EGTA, and adjusted to pH 7.2. The standard extracellular solution for the whole-cell voltage clamp contained (in mM) 140 NaCl, 2 KCl, 2 $MgCl_2$, 2 $CaCl_2$, and 10 HEPES, and adjusted to pH 7.2. The ion selectivity internal pipette solution for the whole-cell voltage clamp contained (in mM) 1 NaCl, 1 KCl, 2 $CaCl_2$, 2 $MgCl_2$, 110 N-methyl D-glucamine, 10 CHES, and 10 EGTA, and adjusted to pH 9.0. The ion selectivity extracellular solution for the whole-cell voltage clamp contained (in mM) $Ex_{NMG, 9.0}$ is 1 NaCl, 1 KCl, 2 $CaCl_2$, 2 $MgCl_2$, 140 N-methyl D-glucamine, and 10 CHES, and adjusted to pH 9.0. $Ex_{NMG, 6.85}$ is 1 NaCl, 1 KCl, 2 $CaCl_2$, 2 $MgCl_2$, 140 N-methyl D-glucamine, and 10 MES, and adjusted to pH 6.85. $Ex_{NaCl, 9.0}$ is 140 NaCl, 1 KCl, 2 $CaCl_2$, 2 $MgCl_2$, and 10 CHES, and adjusted to pH 9.0. $Ex_{KCl, 9.0}$ is 1 NaCl, 140 KCl, 2 $CaCl_2$, 2 $MgCl_2$, and 10 CHES, and adjusted to pH 9.0. $Ex_{CsCl, 9.0}$ is 1 NaCl, 1 KCl, 140 CsCl, 2 $CaCl_2$, 2 $MgCl_2$, and 10 CHES, and adjusted to pH 9.0. $Ex_{CaCl2, 9.0}$ is 1 NaCl, 1 KCl, 70 $CaCl_2$, 2 $MgCl_2$, and 10 CHES, and adjusted to pH 9.0. $Ex_{MgCl2, 9.0}$ is 1 NaCl, 1 KCl, 2 $CaCl_2$, 70 $MgCl_2$, and 10 CHES, and adjusted to pH 9.0. All solutions of pH were adjusted with N-methyl D-glucamine or HCl. The liquid junction potential was calculated and compensated by pClamp 10.6 software. Time constants were determined by a single exponential fit unless noted.

### 2.4. Optics

For whole-cell voltage clamp, irradiation at 470 or 530 or 590 nm was carried out using WheeLED and collimated LED (parts No. WLS-LED-0530-03 or LCS-0530-03-22, WLS-LED-0590-03 Mightex, Toronto, ON, Canada) or an ND YAG flash laser, Mini lite at 532 nm (Continuum, San Jose, CA, USA) controlled by computer software (pCLAMP10.6, Molecular Devices). Light power was measured directly by an objective lens of a microscope by a power meter (LP1, Sanwa Electric Instruments Co., Ltd., Tokyo, Japan).

## 2.5. Confocal Images

Cell images shown in Figure 1A,B were observed by a Nikon A1 LFOV through an objective lens, Apo 60x Oil λS DIC N2.

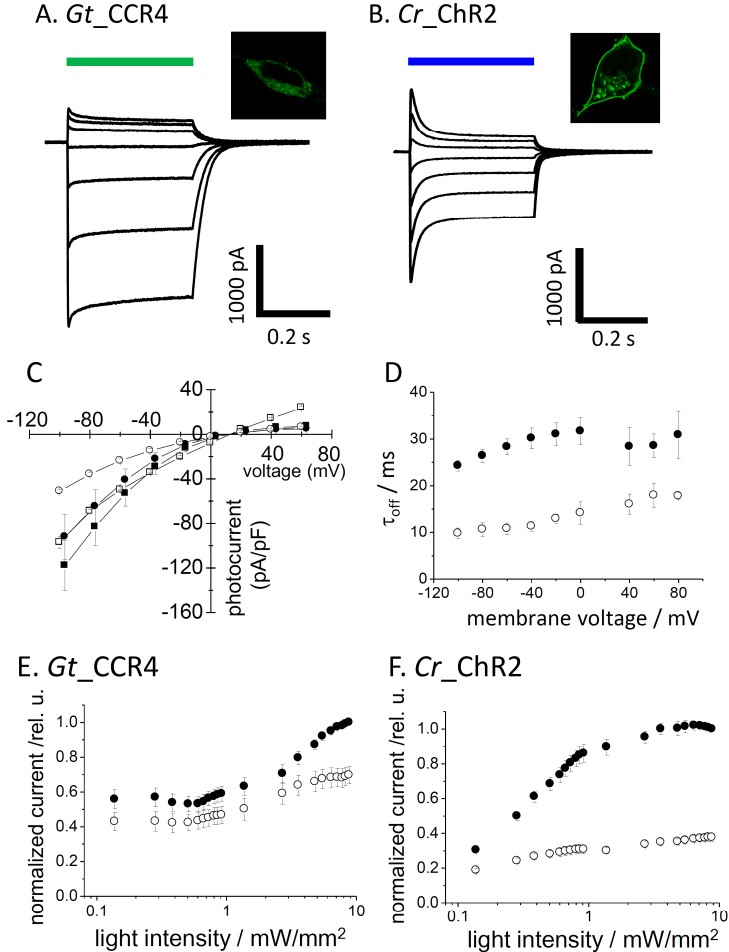

**Figure 1.** Basic properties of *Gt*_CCR4 and *Cr*_ChR2. (**A,B**) Each cation channelrhodopsin expressed in ND7/23 cells was stimulated by green (530 nm) or blue (470 nm) LED light (6.8 mW/mm$^2$). Standard solutions were used. Membrane potentials were clamped from −60 mV to +60 mV in +20 mV steps. Fluorescence images were taken using a confocal microscope. (**C**) Current-voltage relationship (I-V plot) of *Gt*_CCR4 (filled symbol) and *Cr*_ChR2 (empty symbol). Current peak component (square) and steady state amplitude (circle) of two channels are depicted. (**D**) Current-decay kinetics of *Gt*_CCR4 and *Cr*_ChR2. $\tau_{off}$ is plotted as a function of membrane voltage. Filled circle; *Gt*_CCR4, empty circle; *Cr*_ChR2. (**E,F**) Light power dependency of photocurrents from *Gt*_CCR4 and *Cr*_ChR2 at −60 mV. Each channel was stimulated by 530 nm (*Gt*_CCR4) and 470 nm (*Cr*_ChR2). Photocurrent values are normalized. Current peak component (filled circle) and steady state amplitude (empty circle) are depicted. (n = 4–8 cells).

## 2.6. Statistical Analysis

All data in the text and figures are expressed as mean ± SEM.

## 3. Results

### 3.1. Basic Characterization of Photocurrent

We transiently expressed *Gt*_CCR4 and *Cr*_ChR2 in ND7/23 cells by a conventional transfection method (FuGENE). Expression of these channels was visualized by tagged-GFP or Venus fluorescence. Strong membrane expression was confirmed for both channels, although cytoplasmic GFP was observed in *Gt*_CCR4-expressing cells (Figure 1A microscope images). We illuminated 530 nm light to induce a photocurrent for *Gt*_CCR4 and 470 nm light for *Cr*_ChR2 at the same light intensity (6.8 mW/mm$^2$). A large photocurrent was recorded from the *Gt*_CCR4-expressing cells, reproducing previous studies (Figure 1A, left) [30]. Current amplitude reached −2 nA at −60 mV. The current showed an initial peak component ($I_p$) which decayed slightly into a steady state level ($I_{ss}$). However, amplitude of the steady state still retained about 80% of the transient peak component. The photocurrent from *Cr*_ChR2 showed a large peak component which reached about −2 nA at −60 mV (Figure 1A, right). The current decayed by 50% of the initial peak, suggesting that *Cr*_ChR2 exhibits a markedly large inactivation compared to *Gt*_CCR4. Figure 1C depicts the current-voltage relationship of photocurrent from *Gt*_CCR4 and *Cr*_ChR2. Both peak component ($I_p$) and steady state current ($I_{ss}$) are plotted. The shape of the I-V plot from *Gt*_CCR4 indicates strong inward-rectification. $I_{ss}$ of *Cr*_ChR2 displayed similarly inward-rectification, while $I_p$ was weakly rectified, in which a markedly outward-directed current was observed at positive membrane voltages. For both $I_p$ and $I_{ss}$, *Gt*_CCR4 showed a larger current density (pA/pF) than *Cr*_ChR2. For example, the photocurrent ($I_{ss}$) of *Gt*_CCR4 at −100 mV exceeded −80 pA/pF, while that of *Cr*_ChR2 was only about −40 pA/pF.

Kinetics in the photocurrent decay after shutting off the light is shown in Figure 1D. The time constant of *Gt*_CCR4 is about 25–35 ms under a membrane voltage between −100 and 80 mV, while *Cr*_ChR2 showed faster kinetics by about 10–20 ms. Next, we compared light sensitivity in the photocurrent of two channels (Figure 1E,F). The photocurrent amplitude from *Cr*_ChR2 grows as a typical sigmoidal curve for both the initial peak and the steady state components (Figure 1F). $EC_{50}$ was determined as 0.8 mW/mm$^2$ for $I_p$ and 0.35 mW/mm$^2$ for $I_{ss}$ under the conditions tested. On the other hand, the *Gt*_CCR4 current showed unique growth in terms of power dependency with two apparent phases in which the current first saturated at 0.1 mW/mm$^2$ at about 50% of full activation, followed by the second phase of growth from 1 to 10 mW/mm$^2$ (Figure 1E). The $EC_{50}$ was determined as 0.13 mW/mm$^2$ for $I_p$ and 0.18 mW/mm$^2$ for $I_{ss}$. These results indicate that *Gt*_CCR4 is more sensitive to light with respect to channel activation.

### 3.2. Ion Selectivity

It was already reported that *Gt*_CCRs are H$^+$- and Na$^+$-permeable cation channels [29,30]. We here investigated the cation selectivity of *Gt*_CCR4 in more detail relative to *Cr*_ChR2. Ionic conditions of the extracellular solution were systematically exchanged with various cations including Na$^+$, K$^+$, Cs$^+$, Ca$^{2+}$, Mg$^{2+}$ and NMG. In the presence of NMG, one can assume H$^+$ as the permeated ion. Figure 2A,B show the I-V plot of *Gt*_CCR4 and *Cr*_ChR2 under several ionic conditions. Obviously, the reversal potential shift in Na$^+$ solution is larger in *Gt*_CCR4 than in *Cr*_ChR2, suggesting that *Gt*_CCR4 is more permeable to Na$^+$ than *Cr*_ChR2. Notably, I-V plots of *Gt*_CCR4 at pH 6.85 and 9.0 in NMG are almost identical (Figure 2A) whereas a large shift of reversal potential was observed in *Cr*_ChR2 under the same conditions (Figure 2B). These results indicate that *Gt*_CCR4 has less H$^+$ selectivity than *Cr*_ChR2. The photocurrent amplitude of *Gt*_CCR4 and *Cr*_ChR2 at −60 mV under each condition is summarized in Figure 2C,D. The current amplitude of *Gt*_CCR4 was significantly larger in the presence of Na$^+$ and K$^+$ close to −100 pA/pF and in the presence of Cs$^+$ at about −40 pA/pF. In contrast, only a negligible current was observed at low pH, or in the presence of Ca$^{2+}$ or Mg$^{2+}$. This supports the notion that *Gt*_CCR4 is more of a monovalent metal cation selective channel. In addition, we also tested measurements under a competitive environment in which both Na$^+$ and Ca$^{2+}$ were both added to bath solutions (Figure S1). Interestingly, photocurrents by Gt_CCR4 were suppressed at

a higher $Ca^{2+}$ concentration (40 mM), suggesting that $Na^+$ flow is blocked by $Ca^{2+}$. In contrast, such a large difference in current amplitude was not observed under various conditions in the photocurrent from *Cr*_ChR2 (Figure 2D). This is due to low ion selectivity in *Cr*_ChR2 as was reported previously. To assume the selectivity ratio, the reversal potential shift from the condition with NMG at pH 9.0 is depicted in Figure 2E. The shifts ($\Delta E_{rev}$) in *Gt*_CCR4 are larger than in *Cr*_ChR2 for $Na^+$, $K^+$, and $Cs^+$ indicating that *Gt*_CCR4 is selective for monovalent cations but less selective for $H^+$. In the initial study of *Cr*_ChR2, the permeability ratio was calculated based on the current amplitude [8]. The table in Figure 2F summarizes the ratio both for *Gt*_CCR4 and *Cr*_ChR2. $H^+$ permeability for *Cr*_ChR2 is $0.77 \times 10^5$, close to the reported value ($1 \times 10^6$), while that for *Gt*_CCR4 is $2.1 \times 10^4$, indicating about 37-fold less permeability for $H^+$ in *Gt*_CCR4.

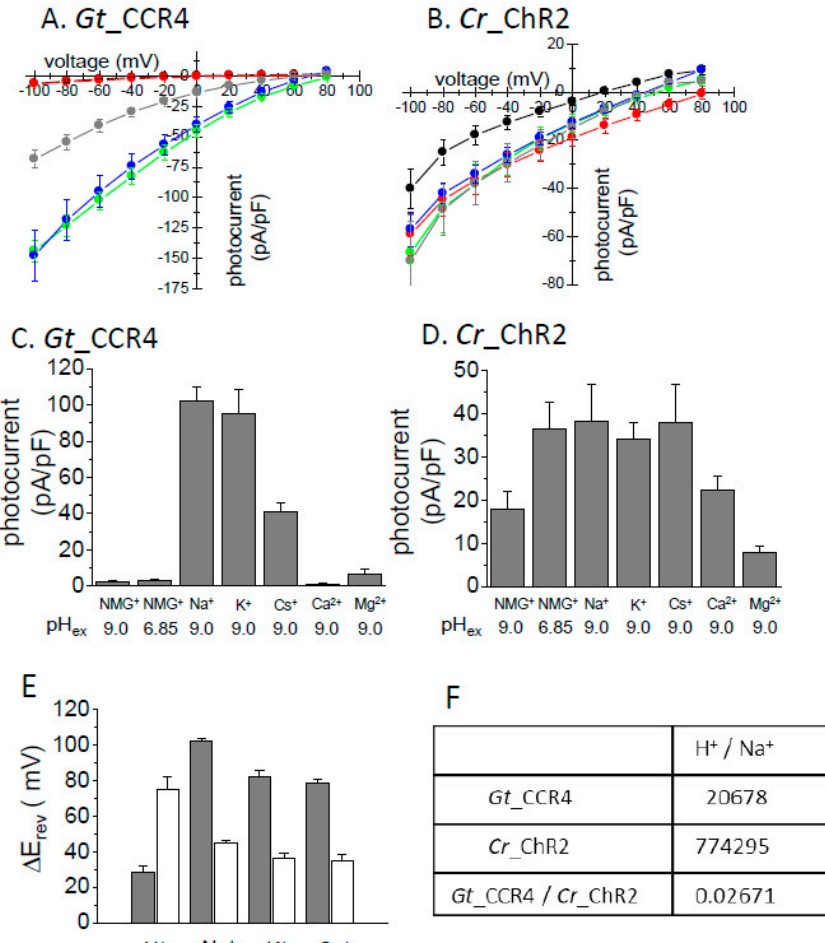

**Figure 2.** Ion selectivity of *Gt*_CCR4 and *Cr*_ChR2. Each channelrhodopsin expressed in ND7/23 cells was stimulated by green and blue LED light. I-V plot of *Gt*_CCR4 (**A**) and *Cr*_ChR2 (**B**) are depicted. Steady-state current density (pA/pF) in 20 mV steps from −100 mV to +80 mV was plotted. The present liquid junction potential was considered. The pipette solution contained 110 mM NMG-Cl at pH 9.0, and the bath solution varied: black, NMG-Cl at pH 9.0; red, NMG-Cl at pH 6.85; green, NaCl at pH 9.0; blue, KCl at pH 9.0; grey, CsCl at pH 9.0. See "Materials and Methods" for details about the solutions. (**C**,**D**) Comparison of current density of *Gt*_CCR4 (**C**) and *Cr*_ChR2 (**D**) in the presence of various cations at −60 mV. (**E**) Reversal potential shift ($\Delta E_{rev}$) for each condition for *Gt*_CCR4 and *Cr*_ChR2. $E_{rev}$ was determined from the I-V plot shown in **A**,**B**. Each $E_{rev}$ value was subtracted from the $E_{rev}$ at NMG-Cl at pH 9.0. (**F**) Permeability ratio for $H^+$ and $Na^+$ in *Gt*_CCR4 and *Cr*_ChR2 as estimated from current value and ionic concentration. (n = 6–9 cells).

### 3.3. Flash Laser Electrophysiology

We then measured the photocurrent under a single-turnover condition with a flash laser as the light source (ND-YAG). As shown in Figure 3A,B, 5 ns light evoked a large inward-directed peak current at −60 mV for both *Gt*_CCR4- and *Cr*_ChR2-expressing cells. The current amplitude and direction are voltage-dependent for both channels, as was expected from recordings with LED. Current growth was fitted with a single exponential function (Figure 3C). The time constant of *Gt*_CCR4 seems to be independent of membrane voltage, while that of *Cr*_ChR2 slowed down slightly as voltage increased. Current decay was determined as about 20 ms for *Gt*_CCR4 and about 5–10 ms for *Cr*_ChR2, both of which are smaller than the value obtained from the measurement by LED light (Figure 1C), suggesting two distinct open states with different kinetics.

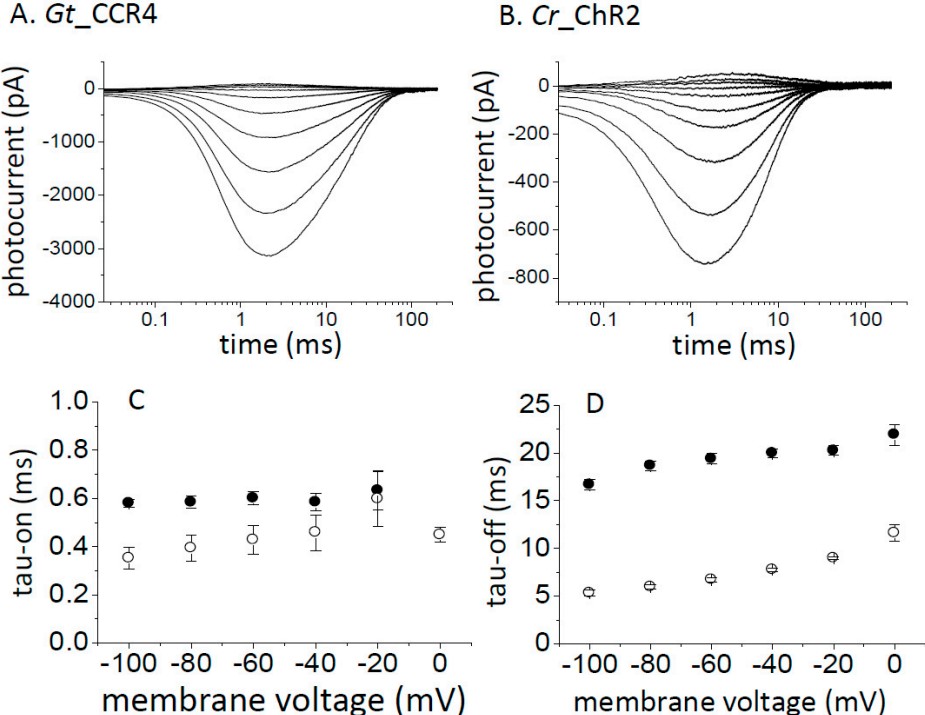

**Figure 3.** Flash laser stimulation. Standard solutions were used. Each channelrhodopsin in ND7/23 cells was stimulated by 5 ns by a green flash laser. Representative trace generated by *Gt*_CCR4 (**A**) and *Cr*_ChR2 (**B**) in 20 mV steps from −100 mV to +80 mV. (**C**) $\tau_{on}$-voltage relationship from *Gt*_CCR4 (filled circle) and *Cr*_ChR2 (empty circle). Current rise was fitted by a single exponential function. The time constant was plotted. (**D**) $\tau_{off}$-voltage relationship from *Gt*_CCR4 (filled circle) and *Cr*_ChR2 (empty circle). Current decay was fitted by a single exponential function. (n = 5–7 cells).

### 3.4. High Frequency Stimulation

For optogenetics application, reliable neuronal activation would require large and stable activity of the light-gated ion channel. In addition, rapid channel-closing is desired for optical stimulation at a high frequency. Thus, we compared the photocurrent from *Gt*_CCR4- and *Cr*_ChR2-expressing cells with three different light frequencies. As shown in Figure 4A, activation of *Gt*_CCR4 at 10 Hz light at 9.67 mW/mm$^2$ generated high temporal peak currents which almost fully decayed before the next illumination. Peak amplitude remained unchanged because of a small inactivation, as shown in Figure 1A. On the other hand, peak current amplitude immediately decayed less after the initial stimulation in *Cr*_ChR2-expressing cells (Figure 4B). Stimulation of *Gt*_CCR4 at 20 and 50 Hz still retained a high level of peak amplitudes, although each peak did not decay completely (Figure 4C,E). Current inactivation was observed in *Cr*_ChR2-expressing cells at each frequency (20 and 50 Hz)

(Figure 4D,F). Figure 4G summarizes the residual current level before the next stimulation at each light frequency. At 10 Hz, both *Gt*_CCR4 and *Cr*_ChR2 showed a very low level of the current level, indicating that channels were almost shut off. As frequency increased to 20 Hz and 50 Hz, significant residual currents were observed which exceeded 50% in *Gt*_CCR4 at 50 Hz whereas *Cr*_ChR2 showed a slightly lower residual current, probably because of faster channel kinetics.

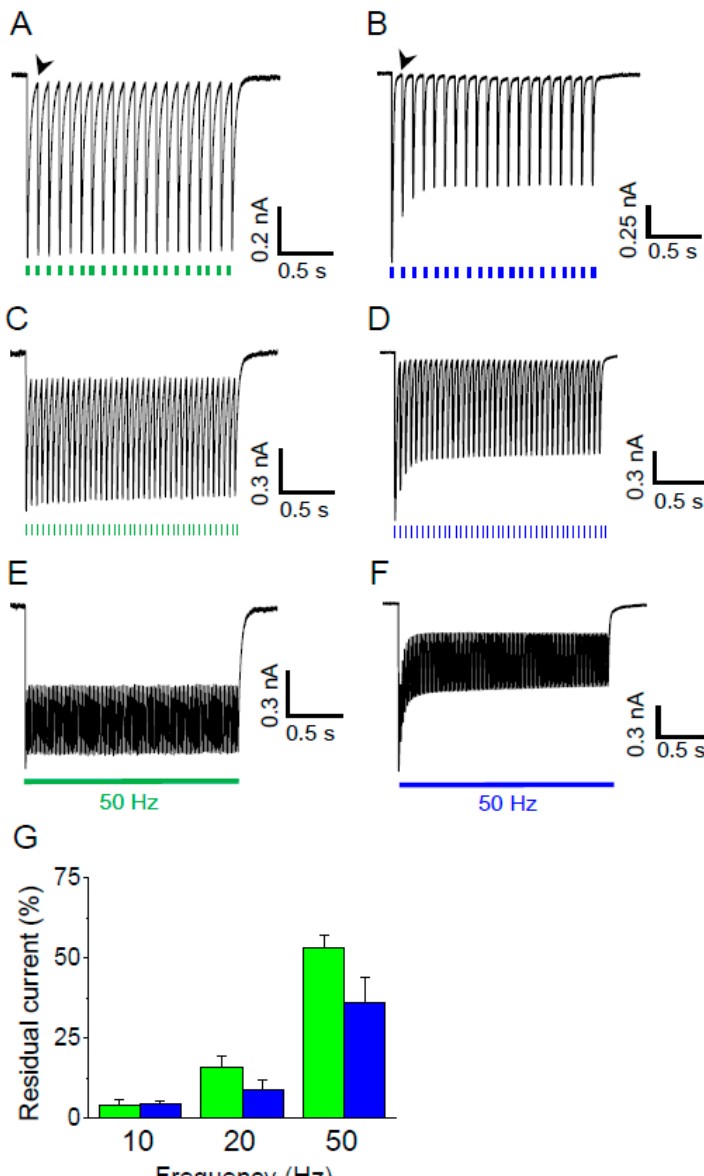

**Figure 4.** High frequency stimulation of *Gt*_CCR4 and *Cr*_ChR2. (**A,C,E**) *Gt*_CCR4 in ND7/23 cells was stimulated by green LED light with a frequency of 10, 20 or 50 Hz as indicated by colored dots or a line under each trace. Membrane voltage was clamped at −60 mV. (**B,D,F**) Similarly, *Cr*_ChR2 was stimulated by blue LED light with a frequency of 10, 20 or 50 Hz as indicated by colored dots or a line under each trace. (**G**) Residual tail current at three different light frequencies is summarized. Residual tail current is the current amplitude right before the next photo stimulation indicated by an arrowhead in **A,B**. The current value was normalized to the peak amplitude as 100%. A standard solution was used. (n = 3 cells).

### 3.5. Gt_CCR4 Is Inactivated by 590 nm Light But Fully Reactivated by Blue Light

As already shown in Figure 1A, one of the obvious characteristics of *Gt*_CCR4 is the small inactivation upon 530 nm illumination which is close to $\lambda_{max}$ (525 nm). Here we compared the current shape by illumination of 530 nm and 590 nm light (Figure 5A–C). Repetitive illuminations of 530 nm light three times gave almost the same current shape, which has a high steady state level (Figure 5A). Upon illumination of 590 nm light, the current slowly decayed into a small steady state level, which was further reduced after a second illumination of 590 nm LED following several seconds of a dark period (Figure 5B,D). Even after 30 sec or 2 min, the inactivated current did not recover (Figure S1A,B). These observations suggest that *Gt*_CCR4 possesses a long-lived inactivated state which could accumulate in 590 nm light. Only illuminating with 530 nm allowed for a full recovery to the original steady state level (Figure 5C,E).

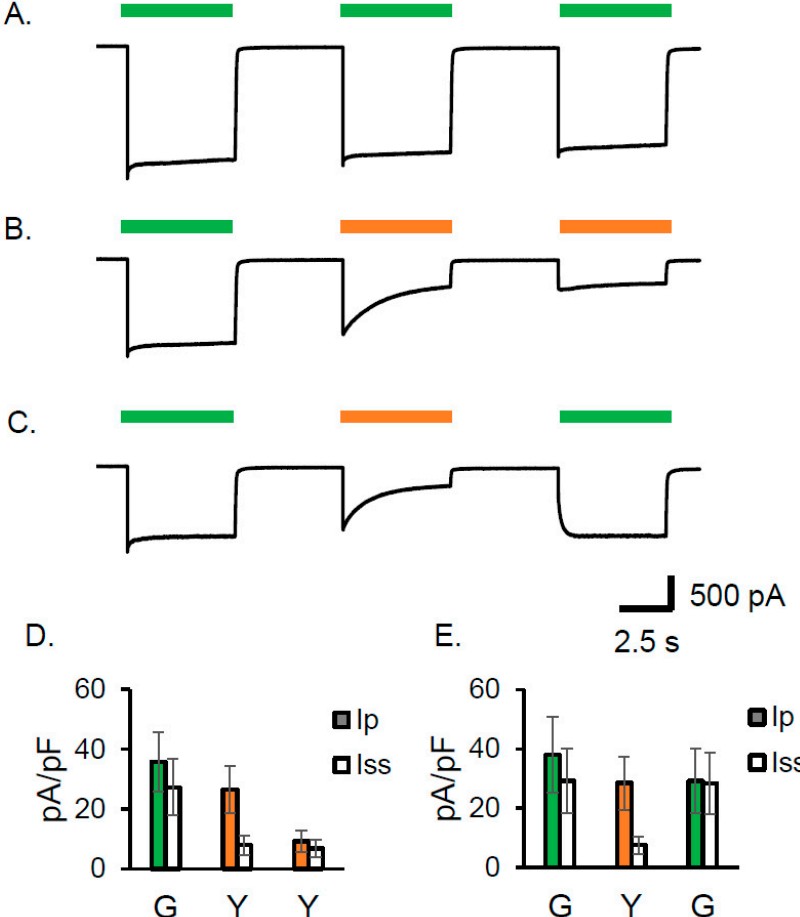

**Figure 5.** *Gt*_CCR4 is inactivated by light with a longer wavelength. Standard solutions were used. Membrane voltage was clamped at −60 mV. Photocurrent by 530 nm light (7.44 mW/mm$^2$) in (**A**) reached a steady state level after transient peak current. Repetitive stimulations gave almost identical current shapes. (**B**) Slow inactivation of the photocurrent was observed by illumination with 590 nm light (7.44 mW/mm$^2$). The current was further reduced when illuminated twice with 590 nm light. (**C**) After 590 nm inactivation, 530 nm light fully reactivated the photocurrent to the original steady state level. (**D**) Photocurrent density from the measurement shown in (**B**). The photocurrent density upon exposure to the first 530 nm light (shown in G), and the second and third 590 nm light (Y), is shown. Ip; transient peak component. Iss; steady state component. (**E**) Photocurrent density from the measurement shown in C. The photocurrent density upon exposure to the first 530 nm light (shown in G), the second 590 nm light (Y), and the third 530 nm light (shown in G), is shown. (n = 4 cells).

## 4. Discussion

In this study, we aimed to elucidate the ion-channel properties of a recently discovered light-gated cation channel *Gt*_CCR4 from a cryptophyte and compare it to well-known *Cr*_ChR2 from *Chlamydomonas reinhardtii* by using an electrophysiological method. In ND7/23 cells, *Gt*_CCR4 showed a large current density (Figures 1C and 2A,B). Inactivation of the photocurrent obtained from *Gt*_CCR4 was smaller than in *Cr*_ChR2. In other words, a large current was observed under constant light (Figure 1A,B). This was also obvious in the current trace after illumination at a high frequency (Figure 4). These characteristics promise stable and reproducible stimulation of neuronal excitability by *Gt*_CCR4.

The light sensitivity of *Gt*_CCR4 is higher than that of *Cr*_ChR2 with a particular steady state component ($I_{ss}$) (Figure 1E,F). ChR variants with high light sensitivity have already been developed [19,20], but those have a long channel life time with at least two orders of magnitude or even much longer. Therefore, these are inappropriate for high-frequency light stimulation. In contrast, *Gt*_CCR4 has a short open life time of 25–30 ms, which is about the same range as *Cr*_ChR2, i.e., 10–15 ms (Figure 1D). Together, *Gt*_CCR4 is light sensitive and useful as an optogenetics tool with high time resolution. Optical irradiation causes heat and elevates temperature by 0.2~2 °C, especially in cranial nerve experiments [37]. Moreover, it has been demonstrated that the rise in temperature suppressed neuronal spiking in multiple brain regions, serving as a warning of the use of strong light for neuronal stimulation. Such an undesirable artefact has to be avoided by lowering light intensity, while effective depolarization has to be stably maintained. *Gt*_CCR4 has the potential for overcoming this problem.

$H^+$ permeability is high for *Cr*_ChR2 [8]. Permeability for $Ca^{2+}$ has been reported, and not only for monovalent cations such as $Na^+$ and $K^+$ [8,21]. On the other hand, *Gt*_CCR4 showed high selectivity in monovalent metal cations and low $H^+$ permeability. The permeability of a divalent cation such as $Ca^{2+}$ seems to be very low or negligible. The position that is important to ion selectivity has been studied in *Cr*_ChR2. E90 in the central gate is crucial for cation/anion selection [23]. L132 in TM3 influences on $Ca^{2+}$ permeability [21]. Duan and coworkers recently demonstrated that D156H and D156C mutation increase permeability for $Na^+$ and $K^+$ [38]. The outer gate in Chrimson (E139) on the extracellular side is important for $Na^+$ extrusion [22]. These key residues are not conserved in *Gt*_CCR4, implying that a different ion selection property resides in DTD channels. It would be necessary to study selectivity based on variant analysis and structural information in the future. Considering its application in optogenetics, *Gt*_CCR4 would not cause a significant change to pH in the cell membrane because of its very low $H^+$ permeability, which could be advantageous when an unknown effect by pH needs to be prevented. To enable optical stimulation without improper calcium signaling, *Gt*_CCR4 might work better than *Cr*_ChR2.

A single turnover photocurrent of *Gt*_CCR4 by laser irradiation provided a time constant ($\tau_{off}$) of 15–20 ms, which is smaller than that obtained by constant light (25–30 ms). This suggests two processes for channel opening and shutting. A dual photocycle model was indeed proposed for *Cr*_ChR2 [39]. It is expected that a similar reaction is caused in *Gt*_CCR4, but more experiments are needed to prove this.

We found a characteristic inactivation of channel activity by long wavelength absorption in *Gt*_CCR4 (Figure 5). Since inactivation lasted at least a few minutes, formation of a stable intermediate with long wavelength absorption is anticipated. Alternatively, *Gt*_CCR4 exhibits a photochromic property that is seen in the photocycle of *Anabaena* sensory rhodopsin [40]. Such photochromism or desensitization was also observed in chlorophyte channelrhodopsins [41,42]. We are now focusing on understanding the reaction mechanism in greater depth via a spectroscopic experiment. In conclusion, we here elucidated the cation channel properties of *Gt*_CCR4. Its high conductance and cation selectivity without significant inactivation would be an appropriate set of features for optogenetics applications. We are currently assessing the feasibility of *Gt*_CCR4 as an optical stimulator in cultured neurons.

**Supplementary Materials:** The following are available online at http://www.mdpi.com/2076-3417/9/17/3440/s1, Table S1 Amino acid alignments of bacteriorhodopsin (BR), *Cr*_ChR2 and *Gt*_CCR4. Figure S1: Ca$^{2+}$ photocurrents of *Gt*_CCR4 and *Cr*_ChR2. Figure S2 Gt_CCR4 has a long-lived and long wavelength-absorbing inactivated state.

**Author Contributions:** Conceptualization, All authors; Data Curation, S.S. and S.H.; Validation, S.S., S.H. and S.P.T.; Writing—Original Draft Preparation, S.P.T.; Writing—Review & Editing, S.P.T.; Supervision, H.K. and S.P.T.

**Funding:** This work was funded by the Japanese Ministry of Education, Culture, Sports, Science and Technology (25104009, 15H02391 to H.K and 18K06109 to S.P.T.), a JST CREST grant (JPMJCR1753 to H.K.), and a JST PRESTO grant (JPMJPR1688 to S.P.T). S.H. is a Research Fellow of the Japan Society for the Promotion of Science (JSPS Research Fellow).

**Acknowledgments:** We thank Ryoko Nakamura for her excellent technical support.

**Conflicts of Interest:** The authors declare no conflict of interest.

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
