# Peer review of "Ion Channel Properties of a Cation Channelrhodopsin, Gt_CCR4"

_applsci, doi:10.3390/app9173440_

Round 1

Reviewer 1 Report

Shigemura et al. here described the characterization of  Gt_CCR4 ion channel properties. This is of interest because channelrhodopsins with distinct ion permeabilities as optogenetic tools would help to learn more detail about cell physiology. My major concern about this manuscript is that some conclusions of the paper is untenable according to previous publications by others.

Important points are:

Line 27-29, with only the comparison between Gt_CCR4 and Cr_ChR2 in this manuscript, it can not suggest “that ion selectivity in DTD channelrhodopsin is distinct from that in chlorophyte channelrhodopsins.” Furthermore, Govorunova et al. already published chlorophyte channelrhodopsins (PsChR and MvChR) with high sodium conductance and low proton conductance in 2013. J Biol Chem 288(41)

Line 309-310, “The position that is important to ion selectivity has already been identified in Cr_ChR2.” In fact, the important positions for ion selectivity are not very clear since Cr_ChR2 is a complicated channel with several gates from inside to outside. Besides E90 in TM2, L132 in TM3 also influence the ion selection, Duan et al in Appl. Sci. 20199(4), 664 found that D156 in TM4 also influence the ion selectivity greatly.

Line 325-327, this feature is not unique for Gt_CCR4, according to a review fro Lin et al. in 2011. Exp Physiol. 96(1), all known channelrhodopsins desensitize (the so-called inactivated or photochromic property in this manuscript). This issue was also described by Zamani et al. in 2017. Biophys Physicobiol. 14. The Cr_ChR2 also desensitize at longer wavelength (470 nm) and become less desensitize at shorter wavelength (430-440 nm). This is also well studied for Halorhodopsin which desensitize greatly at 590 nm but can be recovered by green or blue light.

I would suggest the authors to rewrite the above related part.

Very important points for figures and data are:

Althought the statistical analysis mentioned in the method part, the statistical significance was never shown in any figure.

Experiment repeats number “n” is never provided in any Figure.

Important mistakes are:

Line 61-62, Mutations at C128 to T, A and S slowed different kinetics, not all are of 1000-fold changing.

Line 62-63, D156C was not mentioned in this ref 30.

Line 87, E156 should be D156.

In Table S1, No.83 in Gt_CCR4 is written as a “Q”, but the Gt_CCR4 sequence in NCBI (ARQ20888) showed that position 83 is a “V”

Minor points

Line 2, The “powerful” in the title is not convincing to me, since the authors only compared with wt Cr_ChR2 and showed 2x higher current. Many more powerful Cr_ChR2 mutants and new channelrhodopsins were published and applied. If the authors want to show that Gt_CCR4 is powerful, then they should also compare with published powerful ones.

Line 35-36, several microbial rhodopsins with eight transmembrane helices were published in recent years. So it is not always seven transmembrane helices.

Line 51-52, “The number of variant molecules has been” to “A number of variant molecules have been”

Line 58, TsChR PsChR to TsChR PsChR

Line 69, “A novel cation channelrhodopsin” to “A novel cation channelrhodopsin family”

Line 96, ref is missing since “was described previously”

Line 163, ref is missing for “previous studies”

Figure 3 C, D text below are not of same size.

Line 298 and 320, ChR2 to Cr_ChR2 for consistence.

Figure S1, please indicate the wavelength and intensity of LED light.

Author Response

Dear reviewer,

Thank you very much for reviewing our MS and valuable comments. Please find answers below.

Comment1 : Line 27-29, with only the comparison between Gt_CCR4 and Cr_ChR2 in this manuscript, it can not suggest “that ion selectivity in DTD channelrhodopsin is distinct from that in chlorophyte channelrhodopsins.” Furthermore, Govorunova et al. already published chlorophyte channelrhodopsins (PsChR and MvChR) with high sodium conductance and low proton conductance in 2013. J Biol Chem 288(41)

Answer 1: The reviewer is correct in the point that ion selectivity in DTD channelrhodopsin is distinct from that in channelrhodopsins form chlorophyte. We correct the text as follows. line 27-28 “These results suggest that ion selectivity in Gt_CCR4 is distinct from that in Cr_ChR2.”

Comment2: Line 309-310, “The position that is important to ion selectivity has already been identified in Cr_ChR2.” In fact, the important positions for ion selectivity are not very clear since Cr_ChR2 is a complicated channel with several gates from inside to outside. Besides E90 in TM2, L132 in TM3 also influence the ion selection, Duan et al in Appl. Sci. 2019, 9(4), 664 found that D156 in TM4 also influence the ion selectivity greatly.

Answer 2: We have modified the text with more details about the previous study in ChR2, as follows. line 307-312 “The position that is important to ion selectivity has been studied in Cr_ChR2. E90 in the central gate is crucial for cation/anion selection (23). L132 in TM3 influences on Ca2+ permeability (21). Duan and coworkers recently demonstrated that D156H and D156C mutation increase permeability for Na+ and K+ (37) . The outer gate in Chrimson (E139) on the extracellular side is important for Na+ extrusion (22). These key residues are not conserved in Gt_CCR4, implying that a different ion selection property resides in DTD channels.”

Comment3: Line 325-327, this feature is not unique for Gt_CCR4, according to a review fro Lin et al. in 2011. Exp Physiol. 96(1), all known channelrhodopsins desensitize (the so-called inactivated or photochromic property in this manuscript). This issue was also described by Zamani et al. in 2017. Biophys Physicobiol. 14. The Cr_ChR2 also desensitize at longer wavelength (470 nm) and become less desensitize at shorter wavelength (430-440 nm). This is also well studied for Halorhodopsin which desensitize greatly at 590 nm but can be recovered by green or blue light.

Answer 3: We have corrected the text as follows, line 326-329. Such photochromism or desensitization was also observed in chlorophyte channelrhodopsins (40, 41). We are now focusing on understanding the reaction mechanism in greater depth via a spectroscopic experiment.

Very important points for figures and data are:

Point 1: Althought the statistical analysis mentioned in the method part, the statistical significance was never shown in any figure.

Answer 1, We are very sorry that the method part include a sentence about statistical analysis which we did not performed. We have deleted the sentence.

Point 2: Experiment repeats number “n” is never provided in any Figure.

Answer 2: Thank you for the suggestion. We have added n in all figures.

Important mistakes are:

Mistake 1: Line 61-62, Mutations at C128 to T, A and S slowed different kinetics, not all are of 1000-fold changing.

Answer 1: We have changed the sentence as follows, line 60-61. Mutations at C128 to T, A and S slowed the kinetics of channel closing 200, 5000 and 10000-fold respectively (19).

Mistake 2: Line 62-63, D156C was not mentioned in this ref 30.

Answer 2: Changed to ref 20.

Mistake 3: Line 87, E156 should be D156.

Answer 3: Corrected.

Mistake 4: In Table S1, No.83 in Gt_CCR4 is written as a “Q”, but the Gt_CCR4 sequence in NCBI (ARQ20888) showed that position 83 is a “V”

Answer 4: Thank you for checking the sequence. After re-alignment between Gt_CCR4 and Cr_ChR2, we noticed that E 90 in Cr_ChR2 is better aligned to N84. This might be due to setting of the alignment software. Therefore we showed No. 84 in the revised Table 1.

Minor points

Point 1: Line 2, The “powerful” in the title is not convincing to me, since the authors only compared with wt Cr_ChR2 and showed 2x higher current. Many more powerful Cr_ChR2 mutants and new channelrhodopsins were published and applied. If the authors want to show that Gt_CCR4 is powerful, then they should also compare with published powerful ones.

Answer 1: Thank you for the comment. The reviewer is correct. We only compared Gt_CCR4 and Cr_ChR2, whereas many other ChR variants with robust activity have been reported. We deleted “powerful” from the title.

Point 2: Line 35-36, several microbial rhodopsins with eight transmembrane helices were published in recent years. So it is not always seven transmembrane helices.

Answer 2: Corrected to “seven or eight” line 34.

Line 51-52, “The number of variant molecules has been” to “A number of variant molecules have been”

Point 3: Line 58, TsChR PsChR to TsChR PsChR

Answer 3: Corrected. Line 57

Point 4: Line 69, “A novel cation channelrhodopsin” to “A novel cation channelrhodopsin family”

Answer 4: Corrected. Line 69

Point 5: Line 96, ref is missing since “was described previously”

Answer 5: Added. Line 96

Point 6: Line 163, ref is missing for “previous studies”

Answer 6: Added. Line 161

Point 7: Figure 3 C, D text below are not of same size.

Answer 7: Corrected

Point 8: Line 298 and 320, ChR2 to Cr_ChR2 for consistence.

Answer 8: Corrected

Point 9: Figure S1, please indicate the wavelength and intensity of LED light.

Answer 9: Indicated

Reviewer 2 Report

The paper is interesting and presents new discovery about a member in BCCR in the field of microbial rhodopsins.

I have four comments.

First, one of the advantages of GtCCR4 is that it has higher selectivity towards Na+ than to H+. Can the authors discuss about the molecular mechanism behind this phenomenon?

Second, the authors presents that E90 from ChR2 and E139 from chrimson are not conserved in GtCCR4. Therefore the residues which are responsible for selection towards cations against anions are not known. Can the authors discuss about any other residue that could be responsible for ion selectivity?

Third, GtCCR4 has little inactivation after light illumination. It is reported that another Cryptophyte CCR has faster inactivation by Disseroth's group. Can the authors discuss about the potential residues that are cause faster inactivation versus slow inactivation?

Fourth, what is the action spectrum and does it match the previously published absorption spectrum of GtCCR4?

Author Response

Dear Reviewer,

Thank you very much for reviewing our MS and valuable comments. Please find answers below.

First, one of the advantages of GtCCR4 is that it has higher selectivity towards Na+ than to H+. Can the authors discuss about the molecular mechanism behind this phenomenon?

Answer 1: It is difficult at the moment to discuss about molecular mechanism of ion selectivity of Gt_CCR4. As we described in line 307-313, key residues for ion selection in Cr_ChR2 are not conserved in Gt_CCR4. In addition the crystal structure is unknown. We are currently working on mutation study of Gt_CCR4 to assess a selectivity filter. We hope to identify and report it in near future.

Second, the authors presents that E90 from ChR2 and E139 from chrimson are not conserved in GtCCR4. Therefore the residues which are responsible for selection towards cations against anions are not known. Can the authors discuss about any other residue that could be responsible for ion selectivity?

Answer 2: We anticipate that some negatively charged residues such as E or D are crucial for ion selection in Gt_CCR4. In TM3, there are two D (D116 and D127) which might be responsible for ion selectivity. But Sineshchekov and coworkers found that D85 and D98 in Gt_CCR2 (corresponding to D116 and D127 in Gt_CCR4) are responsible for the channel gating, not the cation selectivity (PNAS, 114, 9512-19, 2017). Therefore we do not have any experimental clue for ion selectivity. Further mutation study would be needed.

Third, GtCCR4 has little inactivation after light illumination. It is reported that another Cryptophyte CCR has faster inactivation by Disseroth's group. Can the authors discuss about the potential residues that are cause faster inactivation versus slow inactivation?

Answer 3: Thank you for the interesting comment. We noticed the new publication in Science about a red-shifted Cryptophyte CCR called ChRmine, which shows fast and large inactivation. We are comparing the amino acid sequence between ChRmine and Gt_CCR4. Unfortunately we cannot find out potential residues being responsible for inactivation at the moment. Nevertheless, we found several mutants of Gt_CCR4 which exhibit altered inactivation. We are intensively studying characteristics of those mutants. We hope to report about it in future.

Fourth, what is the action spectrum and does it match the previously published absorption spectrum of GtCCR4?

Answer 4: Yes, the action spectrum peaks at 520 nm which agrees with the previous data in Yamauchi et al. Biophys. Physicobiology. 14: 57–66, 2017.

Round 2

Reviewer 1 Report

I suggest paper to be accepted after spell check of English.

Author Response

Dear Reviewer,

Thank you for reviewing the revised MS. As the reviewer suggested. We have made a spell check of English. 

Sincerely Yours,

Satoshi Tsunoda